# Effects of Whole-Body Vibration Exercise on Athletes with Ankle Instability: A Systematic Review

**DOI:** 10.3390/ijerph20054522

**Published:** 2023-03-03

**Authors:** Ana Carolina Coelho-Oliveira, Redha Taiar, Juliana Pessanha-Freitas, Aline Reis-Silva, Luiz Felipe Ferreira-Souza, Luelia Teles Jaques-Albuquerque, Aline Lennertz, Márcia Cristina Moura-Fernandes, Ana Cristina Rodrigues Lacerda, Vanessa A. Mendonça, Borja Sañudo, Adérito Seixas, François Constant Boyer, Mario Bernardo-Filho, Amandine Rapin, Danúbia Sá-Caputo

**Affiliations:** 1Programa de Pós-Graduação em Fisiopatologia Clínica e Experimental, Universidade do Estado do Rio de Janeiro, Rio de Janeiro 20551-030, RJ, Brazil; 2Laboratório de Vibrações Mecânicas e Práticas Integrativas, Departamento de Biofísica e Biometria, Instituto de Biologia Roberto Alcântara Gomes, Policlínica Universitária Piquet Carneiro, Universidade do Estado do Rio de Janeiro, Rio de Janeiro 20950-003, RJ, Brazil; 3MATériaux et Ingénierie Mécanique (MATIM), Université de Reims Champagne Ardenne, F-51100 Reims, France; 4Mestrado Profissional em Saúde, Medicina Laboratorial e Tecnologia Forense, Universidade do Estado do Rio de Janeiro, Rio de Janeiro 20950-003, RJ, Brazil; 5Programa de Pós Graduação em Ciências Médicas, Universidade do Estado do Rio de Janeiro, Rio de Janeiro 20550-170, RJ, Brazil; 6Programa de Pós-Graduação em Reabilitação e Desempenho Funcional, Faculdade de Ciências Biológicas e da Saúde (FCBS), Universidade Federal dos Vales do Jequitinhonha e Mucuri—UFVJM, Diamantina 39100-000, MG, Brazil; 7Departamento de Educación Física y Deporte, Universidad de Sevilla, 41013 Seville, Spain; 8Escola Superior de Saúde Fernando Pessoa, Fundação Fernando Pessoa, 4200-253 Porto, Portugal; 9Faculté de Médecine, Université de Reims Champagne Ardennes, UR 3797 VieFra, F-51097 Reims, France

**Keywords:** chronic ankle instability, mechanical vibration, vibrating platform, ankle rehabilitation, athletes

## Abstract

Objective: Chronic Ankle Instability (CAI) or Functional Ankle Instability (FAI) is a condition characterized by laxity and mechanical instability in the ankle joint. This instability interferes with the activities and physical-functional parameters of athletes, which leads to repetitive ankle sprains. The current systematic review was carried out to identify the effects of whole-body vibration exercise (WBVE) in athletes with CAI. Methods: We conducted electronic searches in Pubmed, the Cochrane Library, Embase, Web of Science, Scopus, Science Direct, Allied Health Literature (CINAHL) and Academic Search Premier (ASP) (EBSCO) databases on 26 February 2022. Registers were identified, and studies were selected for inclusion according to the eligibility criteria. The methodological quality was assessed by the Physiotherapy Evidence Database (PEDro) scale. Results: Seven studies were included with a mean methodological quality score of 5.85, considered ‘regular’ quality on the PEDro scale. WBVE interventions in athletes with CAI showed that this exercise contributes to a better response on parameters of neuromuscular performance, muscle strength and consequently in balance and postural control, variables that are for the management of CAI. Conclusion: WBVE interventions in sports modalities promote physiological responses that may lead to positive effects in several parameters. The protocols proposed in each modality can be carried out in practice and are considered effective additional exercise and training methods beyond traditional types of training for athletes. However, more studies are needed on athletes with this condition, with specific protocols, to highlight the possible physiological and physical-functional responses. Protocol study registration: PROSPERO (CRD42020204434).

## 1. Introduction

Ankle injuries are common, especially in physically active individuals, and the second most injured part of the body during sports practice is the complex of the ankle joint. Indeed, ankle sprains are one of the most common musculoskeletal injuries among athletes, representing approximately 25–30% of all sport-related injuries [1,2]. The high incidence of acute ankle sprains varies quote by sport, with the highest rates usually reported in sports that involve running, cutting, and jumping, such as football, soccer, basketball, and volleyball [3,4]. This is mainly related to the biomechanics of sports movements, which involve jumping, running, and changing direction [5]. Additionally, Pietro et al. have shown that it is possible that the surface of artificial grass soccer fields contributes to causing more injuries than the natural grass surface in amateur footballers [6].

The ankle functions as a joint complex with cooperation from the talocrural, subtalar, and inferior tibiofibular joints. Typically, lateral ankle sprains occur when the rearfoot undergoes excessive supination with an externally rotated lower leg [7]. Studies suggest that about 73% of individuals affected by ankle sprains may develop residual physical disabilities characterized by persistent symptoms and features, including acute pain, swelling, muscle weakness, loss of range of motion (ROM), deficits in postural control, and joint-sagging sensations. Consequently, there is an increase in recurrent injury with the development of chronic ankle instability (CAI), which denotes the occurrence of repetitive bouts of lateral ankle instability, promoting various ankle sprains [7,8,9,10].

CAI may be due to mechanical instability, functional instability, or, most likely, a combination of these two phenomena. Mechanical instability may be due to specific insufficiencies such as laxity, arthrokinematic changes, synovial inflammation, or degenerative changes. Functional instability would be caused by insufficiencies in neuromuscular control and proprioception. [2,11].

CAI results from a combination of deficits in strength, proprioception [12,13], and neuromuscular control [14], including increased peroneal reaction time [15], reduced activity of the tibialis anterior and peroneus longus [16], an impaired sense of joint position and range of motion [9,10], and decreased balance, and postural stability [13,17,18].

In this context, different exercises that can yield benefits in terms of disabilities due to CAI in athletes have been investigated in this clinical condition, including rehabilitation programs with balance training [19,20,21], strength [20,22], stimuli mechanical vibration, as systemic vibratory therapy (SVT), [10,23,24] and combined training [25,26,27]. However, the evidence is conflicting, and there remains a gap in the literature regarding which type of intervention is most effective and which intervention should be performed after an ankle injury, specifically in athletes, to enable a safe return to sports activity.

SVT promotes the whole-body vibration exercise (WBVE) whereby the individual is in contact with the base of a vibrating platform (VP), which generates mechanical vibrations that are transmitted to the body of the individual. Several parameters must be clearly defined, such as the frequency, amplitude, peak-to-peak displacement, work time, rest time, periodicity of the sessions, and positioning on the VP, amongst others [28,29,30,31].

During WBVE, the mechanical vibrations transmitted by the VP are thought to interact with skin receptors, muscle spindles and joint mechanoreceptors, activating alpha motoneurons that lead to muscle contractions [32,33]. These contractions promoted by WBVE can influence joint stability. Consequently, improvements can be observed in sensorimotor deficits involving balance, joint position sense, and dynamic postural control, with greater brain activity, strength, power, flexibility, and adaptations in motor control, even in populations with joint instability [34,35,36,37,38,39,40,41,42]. Furthermore, in studies with SVT, as a strategy of intervention for athletes, different postures of the individuals and exercises in the base of VP were considered, ranging from one-legged and two-legged exercises, with and without heel lift, involving balance and stability [34,35,36,37,38,39,40,41,42].

In this context, WBVE is a biophysical modality that provides systemic mechanical vibration signals through mechanical stimulation and is an exercise strategy that is gaining popularity and is being increasingly used as a preventive and rehabilitation tool. Currently, there is only one systematic review about WBVE and sensorimotor effects that supports the use of WBVE to improve sensorimotor deficits involving strength, balance, muscle activity, and joint position sense, but it was in individuals with CAI, not athletes [43]. Considering this rationale, the idea is to present the possibility of using an efficient exercise program to assist in the dysfunctions promoted by CAI, which is useful to stimulate the physical-functional performance of the athlete, as well as the return to sports practice. So, the aim of this study was to present a systematic review to identify the effects of WBVE in athletes with CAI.

## 2. Materials and Methods

The review was realized according to the Preferred Reporting Items for Systematic Reviews and Meta-Analysis (PRISMA, Appendix A) guidelines [44] and the Synthesis without meta-analysis (SWiM) in systematic reviews [45]. The protocol for the review was registered with the International Prospective Register of Systematic Reviews (PROSPERO—CRD42020204434).

### 2.1. Search Strategy

An electronic search in databases was realized in Pubmed, Cochrane Library, Embase, Scopus, Science Direct, Web of Science, Allied Health Literature (CINAHL) and Academic Search Premier (ASP) (EBSCO) on 3 September 2021 and updated on 26 February 2023, using the following search strings: (“whole body vibration” OR “vibration therapy” OR “vibration training”) AND (“chronic ankle instability” OR “ankle instability” OR “functional ankle instability” OR “mechanical ankle instability” OR “recurrent ankle instability”). Using the PICOS strategy, the keywords used in the search were defined, such as athletes with CAI (Participants) receiving WBVE intervention (Intervention), allowing comparison to control/no intervention, placebo or usual care (Comparison). All reported outcomes related to balance, muscle strength, muscle activity and body composition (Outcomes) were allowed if they were considered relevant to the population studied and were “randomized clinical trials” (RCT) (Study design) [46]. Additionally, a hand search was performed in gray literature and in studies references.

### 2.2. Eligibility Criteria

Inclusion criterion: RCT, studies with the use of WBVE in athletes with ankle instability, independently of the year of the publication, and with no restrictions on language.

Exclusion criteria: Congress abstracts, incomplete articles, abstracts, pilot studies, protocols, postintervention results not reported, and findings not related to CAI and athletes and WBVE.

### 2.3. Level of Evidence of the Selected Articles

The level of evidence of the included studies was evaluated using the National Health and Medical Research Council hierarchy of evidence [47] and defined according to below Table 1.

### 2.4. Methodological Quality and Risk of Bias

The Physiotherapy Evidence Database (PEDro)scale (http://www.pedro.org.au/english/downloads/pedro-scale/ (accessed on 5 April 2022)), with eleven items, was used to the methodological quality of the randomized controlled trials included. The first item of the PEDro scale is related to external validity and is not used to calculate the scale score. Therefore, the scores range from 0 to 10. Articles with a score greater than or equal to 7 were of ‘high’ methodological quality, those with a score of 5 to 6 were of ‘regular’ quality, and those with a score of 4 or less were of ‘poor’ quality [48,49].

The Cochrane Collaboration tool, with 7 domains, was used to assess the risk of bias in the articles included. The domains encompass selection, performance, detection, attrition, reporting, and other sources of bias. The domains are classified into 1 of 3 categories: low, high or uncertain risk of bias [50].

### 2.5. Selections and Data Extraction

All references found by the database searches were exported to a data management software (EndNote X9 (Clarivate Analytics, London, UK)), and duplicates were removed. Studies were independently reviewed by 2 authors (A.C.C.-O. and J.P.-F.) according to the eligibility criteria at all stages of the review. If disagreements could not be resolved through discussion, a third reviewer (DC) was consulted. For data extraction, both reviewers extracted the following information: study information (author and year), study design, subjects (sample size), demographics characteristics (age, sex, body mass index), modality of athletes, instruments, tests/measured variables, WBVE intervention, WBVE protocols, and outcomes.

## 3. Results

A total of 135 records were identified through database searches (Pubmed = 14, Cochrane Library = 22, Embase = 16, Scopus = 18, Science Direct = 22, Web of Science = 24, CINAHL = 8 and ASP (EBSCO) = 11) and, after the removal of 76 duplicates, 59 studies were identified. During the screening process, the titles and abstracts of 59 publications were read, and 47 were excluded because they were abstracts, reviews (systematic, scoping or narrative), and meta-analyses, they did not meet the topic criteria, were with individuals or patients with CAI and did not use vibratory therapy. The complete text of 12 publications was reviewed in detail. After careful analysis, five studies were excluded because they were not athletes with CAI and because they did not contain available results. Finally, seven articles were included in the current systematic review. The selection process is shown in Figure 1.

The publications included had a mean methodological quality score of 5.85 as assessed, with the PEDro scale (Figure 2), with a minimum of 3 points and a maximum of 7, reflecting ‘regular’ methodological quality. Only in one study [51] was the allocation of groups performed blindly, and no included study was able to blind all subjects or therapists who administered the therapy; two studies [40,51] applied the blinding of all assessors who measured at least one key outcome, three studies [10,23,36] did not provide at least one key result; only one study [23] failed to provide measurements of all outcomes.

For the risk of bias (Figure 3), among the studies included in this review, only one study [51] guaranteed that it performed allocation concealment, five studies [10,23,26,40,51] did not involve blinding of participants and personnel, two studies [24,36] presented insufficient information to judge this domain. Two studies [40,51] blinded the evaluation of the results, all studies included had no missing data, and all reported the results of the variables analyzed. However, according to the Cochrane classification tool, all the studies included had the potential for other biases due to methodological flaws. The articles included in this review were finally classified as high-risk (two studies [10,23]), low-risk (two studies [40,51]), and uncertain risk (three studies [24,26,36]).

Table 2 displays the characteristics of the articles included, including the author and year of the study, the participants involved, the objectives, the variables evaluated, the results found and the level of evidence. The level of evidence (NHMRC, 2003–2007) [47] of all selected studies was Level of Evidence II.

Regarding the main findings (Table 2), WBVE: improves single-leg balance and SEBT performance in dancers with unilateral functional ankle instability (FAI) [10]; Combined vibration and wobble board training improves COM distribution, modified SEBT scores, and SLTHD among footballers suffering from FAI [23] and may have a beneficial effect on the improvement of lower extremity muscle activity and balance ability in CAI football players [36]. WBVE improved the RT of the PB, PL and TA muscles in CAI recreational athletes [40]; resulted in different enhancements in balance ability on the BBS and SEBT [51]; exhibited very small or small effect sizes for CAI in the SEBT, the joint position sense test, and the isokinetic strength test [26]; and, it appears that the WBT is superior to the conventional PPT program for improving dynamic balance in athletes with FAI while COM and ECC were enhanced at an ankle inversion of 30°/s [24].

The sample sizes ranged from 30 to 63 athletes (totaling 300 athletes, 155 female, 95 male, and 50 unspecified), the average age was 21.5 years and included dancers [10], football [23] or soccer players [36], women’s basketball and volleyball teams [26], physically active recreational athletes [40,51], and collegiate athletes (basketball, football, netball, and handball players) [24], and the conditions varied between unilateral [10,23,40] and bilateral [24,51] ankle instability, and two studies [26,36] did not report specific instability.

The protocols and biomechanical parameters of the WBVE (frequency, peak-to-peak displacement, positioning of the individual at the base of the VP, the VP model and the type of stimulus) are reported in Table 3.

Considering the biomechanical parameters, the vibration frequencies used ranged from 5 to 50 Hz; only one publication defined only the frequency of 5 Hz [26], one study used a progressive frequency from 5 to 25 Hz [33], four studies used frequencies of 30, 35 and 40 Hz [10,23,40,51], and only one article reported frequencies of 30, 40, and 50 Hz [24], on the other hand, the amplitude varied between 2 mm and 8 mm [24,26,36,40,51], and two studies did not report which amplitude was used during vibration [10,23].

Considering the WBVE intervention protocols, in general, the included studies used the VP or other vibration devices with synchronous and alternating vertical stimuli. Four studies used another tool on top of the VP [26,36,40,51], one study performed the exercise directly on the VP [24], and two studies used another vibrating device [10,23]. In five studies, the participants were barefoot [10,23,26,40,51], and in two studies [24,36], they were wearing sneakers. Participants performed different positions and exercises on the platform, ranging from two-legged and one-legged exercises, with and without heel lift, involving balance and stability; the working time involved training lasting from 10 to 30 min, including protocols with progressive training as the exercise weeks progressed.

All studies included were homogeneous with regard to the time of application of the WBVE protocol lasting 6 weeks; however, five studies [24,26,36,40,51] used a frequency of three times a week, totaling 18 sessions, and two [10,23] twice a week, totaling 12 sessions.

## 4. Discussion

The main goal of this systematic review was to identify the effects of WBVE in athletes with ankle instability. After analyzing the studies included and considering their limitations, the results suggest that WBVE may be a feasible and valid intervention for athletes with CAI. The methodological quality of the included studies was regular. However, according to the risk of bias, the included clinical trials have methodological errors and still lack detailed information on the methods used and which may compromise their internal validity.

As observed, among the consequences of CAI, there is a deficit in balance and stability. Five included studies [10,23,24,36,51] suggest that vibration provided additional benefits regarding improvements in unipedal balance, dynamic balance, center of mass distribution (COM), and SEBT, MSEBT, SLTHD, and BBS test performance and scores, even with the authors using different VP devices, different WBVE protocols and parameters, as well as different athlete modalities. Corroborating with these findings, Sofla et al. [52] and Tohidast et al. [53] also suggest that WBVE may induce some improvement in postural control in individuals with CAI using protocols of 4 and 6 weeks of vibration, respectively. On the other hand, Chang et al., 2021 [26] observed very small or small effect sizes for the vibration results in the balance and stability test in athletes. Moreover, Adelman et al., 2016 [54] and Rendos et al. [39] did not observe significant changes with WBVE in terms of balance/stability; however, these studies used an acute effect WBVE protocol (a single session), and the participants were not athletes, which suggests that one session would not be enough to promote changes and adaptations in these parameters regardless of physical activity level. The tests performed in the pre- and post-intervention studies with WBVE are considered indicative of postural control, and balance training improves the individual’s ability to maintain the center of gravity and posture, stimulating the musculoskeletal and vestibular systems. Therefore, stability is the result of the activation of the proprioceptive system and is of fundamental importance for the performance of motor tasks, mainly in the ca of athletes with CAI [24,55].

Regarding muscle strength, Sierra-Guzman et al. [40] and Chang et al. [26] did not find significant results regarding muscle strength of the ankle muscles or found very small or small effects after the intervention, suggesting that the WBVE protocols and parameters applied in the studies (frequency of 30, 35 and 40 Hz and amplitude varying from 2 to 4 mm, and 5 Hz and 3 mm, respectively), may not have promoted sufficient stimuli to provide an increase in the isokinetic strength of the ankle invertor and evertor muscles, even with 6 weeks of training. Similarly, Sofla et al. [52] did not observe significant changes regarding ankle muscle strength in individuals with CAI with a 4-week protocol (frequency 30–40 Hz and 3 mm amplitude) of vibration exercises. In the studies by Sierra-Guzman et al. [40], Chang et al. [26] and Sofla et al. [52], although they used different protocols, exercises and populations, strength was assessed with an isokinetic dynamometer, considered the “gold standard” instrument to measure muscle strength and performance [56]. On the other hand, Cloak et al. [23] observed significant effects on strength with the combined vibration and oscillating board training; however, they used the triple jump test for distance (SLTHD) to assess muscle strength, which is a valid clinical tool to assess strength and power characteristics. The reported differences in ankle, knee, and hip kinematics, kinetics, and muscle activation patterns in individuals with CAI, could be due to those lower limb muscle strength deficits and imbalances, and the proposed WBVE protocols, although progressive and associated with exercises, were not sufficient to balance the musculatures involved to support the sports demand of athletes.

Considering that CAI is one of the commonly observed sports conditions, and given that it causes decreased neuromuscular control and loss of proprioception, with respect to the results for muscle activation, Jeong et al., 2017 [36] suggest that WBVE may have a beneficial effect in improving muscle activity of the lower limbs in soccer players, as well as Cloak et al., 2010 [10] found that WBVE can lead to long term effects in the recruitment of motor units of the peroneus longus in dancers. These findings suggest that effective exercise protocols, independent of the type of vibrating platform, stimulus intensity and exposure time, can be used to increase α motor neuron excitability and motor unit synchronization to increase ankle motor control. In contrast, Sierra-Guzman et al., 2017 [40] found no significant changes in EMG activity after a 6-week WBVE training program on a soft surface, although it could improve ankle muscle RT of the PB, PL and TA muscles in recreational athletes with CAI. Similarly, Otzel et al., 2019 [57] observed changes in motoneuron function with a single session of WBVE, suggesting that both acute and cumulative protocols did not represent a sufficient stimulus to the ankle muscles. Neuromuscular control is therefore important for athletes as it helps maintain functional ankle stability, while proprioception influences the sensation of movement and the ankle joint position.

The strength of the current systematic review lies in offering a possibility for an efficient exercise program to counter the dysfunctions promoted by CAI and useful to stimulate the physical-functional performance of the athlete, as well as the return to sports practice with greater safety and, consequently, less chance of recurrence of the lesion that promoted instability. However, this systematic review has some limitations, and the results should be interpreted with caution. Despite the positive results of interventions with WBVE, the diversity of WBVE training protocols makes it difficult to interpret the findings. On the other hand, there is also a lack of objective evidence to support the use of WBVE in athletes with CAI. In addition, different sports modalities were studied, each with different characteristics which may influence the athlete’s performance. Therefore, it is not possible to draw convincing conclusions based on a small number of relevant studies (only seven publications) evaluating WBVE intervention in the athletic population. In addition, some studies used multiple instruments or tools together with the vibration stimulus, and it was not possible to evaluate the vibration stimulus alone.

In this context, as facts and perspectives, the WBVE is a systemic mechanical vibration intervention capable of stimulating the physical-functional performance of the athletes through efficient exercise to promote benefits in the dysfunctions promoted by CAI. Regarding its prospects, and in view of the limitations of the association of WBVE with other “interventions,” it would be interesting that more studies on systemic vibration, applying WBVE alone, focusing on protocol parameters that would help determine the most efficient and effective protocol to promote beneficial effects in athletes with CAI.

## 5. Conclusions

WBVE interventions in athletes with CAI show that this exercise method improves the response in parameters of neuromuscular performance, muscle strength and, consequently, balance and postural control, which are all indispensable variables for the management of this clinical condition. However, more studies are needed on athletes with this condition, with specific protocols, to highlight the possible physiological and physical-functional responses. The implementation of WBVE interventions in this population, regardless of the sport modality, promotes positive effects in several parameters and is a viable intervention to be carried out in practice. However, more studies are indeed needed to investigate the effects of WBVE in athletes with CAI, as there is evidence that WBVE can be an additional exercise, promising and effective, as a training method beyond traditional types of training for athletes.

## Figures and Tables

**Figure 1 ijerph-20-04522-f001:**
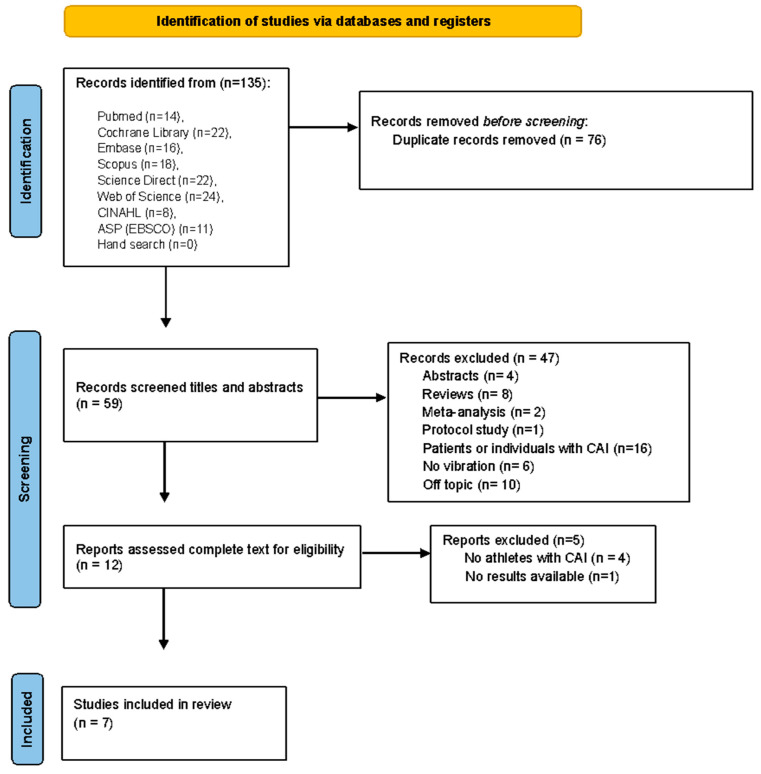
PRISMA flow diagram of the study selection process.

**Figure 2 ijerph-20-04522-f002:**
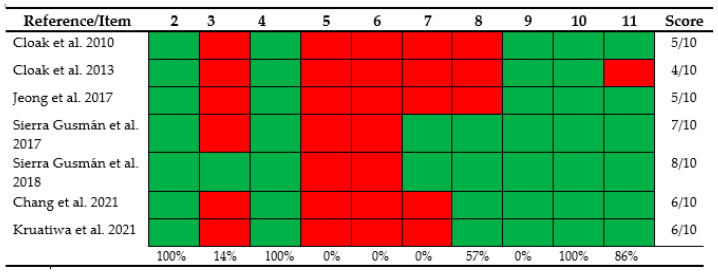
Assessment of the methodological quality of the included studies using the PEDro scale. (2) Subjects were randomly allocated to groups; (3) allocation concealed; (4) the groups were similar at baseline regarding the prognostic indicators; (5) was blinding of all subjects; (6) was blinding of all therapists who administered the intervention; (7) was blinding of all assessors who measured at minimal one key outcome; (8) measures of at least one key outcome were obtained from more than 85%; of the subjects firstly allocated to groups; (9) all subjects for whom outcome measures were disposable received the treatment or control condition as allocated or, where this was not possible, data for at least one key outcome was analyzed by “intention to treat”; (10) the results of between-group statistical comparisons are reported for at minimal one key outcome; (11) the study give both point measures and measures of variability for at minimal one key outcome [10,23,24,26,36,40,51].

**Figure 3 ijerph-20-04522-f003:**
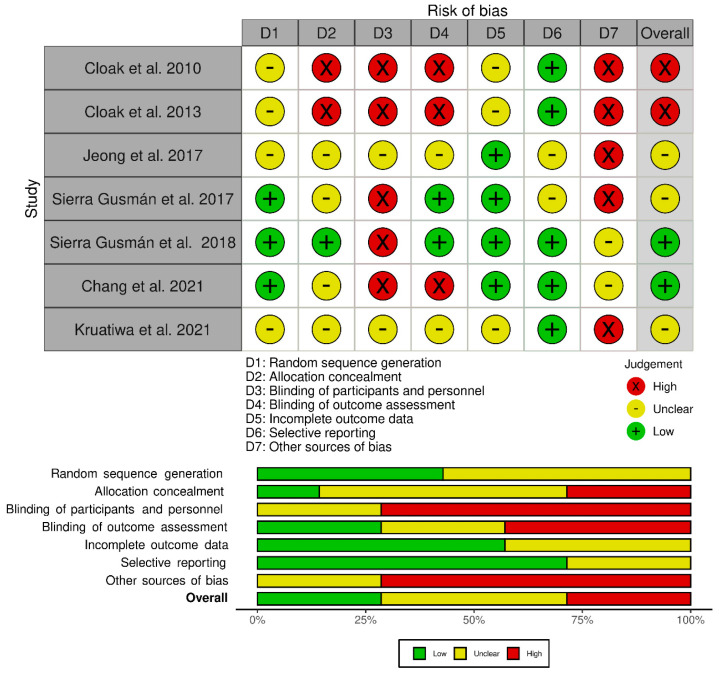
Assessment for risk of bias for the included studies [10,23,24,26,36,40,51].

**Table 1 ijerph-20-04522-t001:** Level Classification.

Level	Classification
I	systematic review of level II studies
II	RCT
III-1	pseudo-randomized or quasi-randomized controlled trial (alternate allocation, such as a crossover study or other similar method)
III-2	a comparative study with concurrent controls (non-randomized trial, case-control study, cohort study, interrupted time series with a control group)
III-3	a comparative study without concurrent control (two or more single-arm studies, historical control, interrupted time series without a parallel control group)
IV	case series with either post-test or pretest/post-test results

**Table 2 ijerph-20-04522-t002:** Table of the characteristics of selected studies.

Study	Study Design	Demographics	Instruments	Variables Measured and Tests	Results	Level of Evidence
Cloak et al., 2010 [10]	Randomized Clinical Trial	n = 38 female dancers (age 19 ± 1.1 years, height 163.6 ± 7.3 cm, body mass 60.3 ± 6.3 kg) WBVT (n = 19) Control groups (n = 19)	RSscan^®^ pressure mat (RSscan, Ipswich); surface EMG measurements with data acquisition system (Powerlab, AD instruments, UK)	Single leg balance test; SEBT; Mean power frequency (fmed) of the peroneus longus.	WBVT improves single-leg balance and SEBT performance in dancers with unilateral FAI. The positive effect of WBVT, its short time of training and its adherence rate in the study supports the need for future research in dance populations, on this type of exercise, as a new method of ankle injury prevention.	II
Cloak et al., 2013 [23]	Randomized Clinical Trial	n = 33 male amateur football players Vibration and wobble board training group (n = 11) age 22.2 ± 0.7, body mass 78.3 ± 7.7 kg, height 174.5 ± 7.8 cm) Wobble board training alone (n = 11) age 22.7 ± 1.2 years, body mass 73.9 ± 4.7 kg, height 171.2 ± 5.4 cm Control group (n = 11) age 23.1 ± 1.1 years, body mass 77.5 ± 7.0 kg, height 176.5 ± 9.0 cm	RSscan pressure mat (RSscan, Ipswich, UK);	COM Distribution; MSEBT; SLTHD to assess the strength	Combined vibration and wobble board training improves COM distribution, MSEBT scores, and SLTHD among footballers with FAI compared with wobble board training alone.	II
Jeong et al., 2017 [36]	Randomized Clinical Trial	n = 30 university soccer players WBVE (n = 15) age 22.51 ± 2.62 years, height 173.65 ± 8.34 cm, body mass 69.89 ± 7.49 kg NMT (n = 15) age 21.94 ± 2.54 years, height 172.74 ± 9.48 cm, body mass 70.57 ± 8.39 kg	Surface electromyography (sEMG) with MP100 surface EMG system (Biopac System Inc., Santa Barbara, CA, USA); Balance measuring system, Biorescue (RM Ingenierie, France)	EMG; Balance ability,	WBVE may have a beneficial effect on the balance ability and improvement of lower extremity muscle activity in CAI football players.	II
Sierra-Gusmán et al., 2017 [40]	Randomized Clinical Trial	n = 50 physically active recreational athletes VIB (n = 17) (age 22.4 ± 2.6 years, height 172.0 ± 8.3 cm, body mass 70.2 ± 8.2 kg) N-VIB (n = 16) (age 21.8 ± 2.1 years, height 171.3 ± 9.0 cm, body mass 66.2 ± 10.1 kg) CON (n = 17) (age 23.6 ± 3.4 years, height 172.7 ± 10.8 cm, body mass 70.6 ± 11.7 kg)	Electromyographic (EMG) (ME6000-T8, Mega Electronics, Kuopio, Finland); Biodex Multi-Joint System 3 dynamometer (Biodex Medical System, New York, NY, USA)	EMG; Isokinetic test in which the ankle evertor muscles strength was tested	The results suggest the intervention with VP could improve the RT of the PB, PL and TA muscles in recreational athletes with CAI. There were no differences in iEMG and isokinetic strength. Taking these results into account, the addition of vibration to balance training could be recommended in order to better the response against a sudden inversion.	II
Sierra-Gusmán et al., 2018 [51]	Randomized Clinical Trial	n = 50 recreational athletes Vibration (VIB = 11 men, 6 women) age 22.4 ± 2.6 years, height 172.0 ± 8.3 cm, body mass 70.2 ± 8.2 kg Nonvibration (NVIB = 10 men, 6 women) age 21.8 ± 2.1 years, height 171.3 ± 9.0 cm, mass 66.2 ± 10.1 kg Control (CON = 12 men, 5 women) age 23.6 ± 3.4 years, height 172.7 ± 10.8 cm, mass 70.6 ± 11.7 kg	Mobile platform interfaced with computer software (version 1.32; Biodex Medical Systems); Dual-energy X-ray absorptiometry (DXA; model Lunar iDXA; General Electric Healthcare, Fairfield, CT, USA).	BBS Test; SEBT; Body-Composition Analysis	The main finding was WBVE training on an unstable surface improved balance. Only the VIB performed better on the BBS, but both training groups performed better on the SEBT. Overall, the results support using balance training with or without WBVE to address balance impairments in participants with CAI. It was also hypothesized that lean mass could be increased with the WBVE training program but observed no change in body-composition variables.	II
Chang et al., 2021 [26]	Randomized Clinical Trial	n = 63 female athletes basketball and volleyball Group A: WBV training program (n = 21) age 20.31 ± 1.28 years, height 168.34 ± 5.78 cm, body mass 61.01 ± 22.39 kg; Group B: balance training program (n =21) age 20.43 ± 1.25 years, height 166.8 ± 6.84 cm, body mass 58.83 ± 13.14 kg; Group C: not training program (n = 21) age 21.23 ± 1.47 years, height 169.53 ± 4.78 cm, body mass 58.67 ± 16.5 4 kg	SYSTEM 3 PRO dynamometer (Biodex Medical Systems, Shirley, NY, USA);	SEBT, A joint position sense test, Isokinetic strength (Test ankle invertor and evertor muscle strength)	Female athletes who participated in the exercise training incorporating a VP and balance ball exhibited very small or small effect sizes for CAI in the SEBT, the joint position sense test, and the isokinetic strength test; in addition, COM and ECC at an ankle inversion of 30°/s were enhanced compared with the control group. It was observed no differences among the variables within the two exercise training programs. A balance training program combining WBVE training with a balance ball may be an available effective strategy for the management of CAI.	II
Kruatiwa et al., 2021 [24]	Randomized Clinical Trial	n = 36 collegiate athletes with FAI (basketball, football, netball, and handball players) PPT Group (n = 12, 6 male and 6 female); age 18.00 ± 3.21 years WBVT Group (n = 12, 5 male and 7 female) age 19.16 ± 2.72 years; Control group (n = 12, 4 male and 8 female) age 19.66 ± 0.98 years.	SYSTEM 3 PRO dynamometer (Biodex Medical Systems, Shirley, NY, USA); program analysis Qualisys Track Manager.	MSEBT; SLDJT	Results demonstrated the efficacy of PPT and WBT programs on dynamic balance in athletes with FAI over a 6-week period. Moreover, it appears that the WBT is superior to the conventional PPT program for improving dynamic balance in athletes with FAI. This information may be used to aid in the reduction and prevention of recurrent ankle sprains in athletes with FAI.	II

WBVE: whole-body vibration exercise; CAI: chronic ankle instability; FAI: functional ankle instability; SEBT: Star Excursion Balance Test; CON: concentric contraction; ECC eccentric contraction; COM: Centre of Mass; SLTHD: single-leg triple hop for distance; SLDJT: Single-Legged Drop Jump; MSEBT: Modified Star Excursion Balance Test; NMT: neuromuscular training; VP: vibrating platform, WBVT: whole-body vibration training; EMG: Electromyographic; RT: reaction time; PB: peroneus brevis; PL: peroneus longus; TA: tibialis anterior; BBS: Biodex Balance System; PPT: proprioceptive training.

**Table 3 ijerph-20-04522-t003:** Protocol for WBVE interventions with selected studies.

Study	WBVE Intervention	Parameters	Type of Vibrating Platform	Positioning	Time WBVE
Cloak et al., 2010 [10]	6 weeks twice a week	Frequency: 30 Hz (1 & 2 weeks), 35 Hz (3 & 4 weeks), and 40 Hz (5 & 6 weeks).	Vibration platform (Bosco, Greece)	2 positions per day: single-leg heel raises and single-leg squats.	Week 1: 3 × 50 s each leg, each position, totaling 10 min Week 2: 3 × 50 s each leg, each position, totaling 10 min Week 3: 3 × 60 s each leg, each position, totaling 12 min Week 4: 3 × 60 s each leg, each position, totaling 12 min Week 5: 3 × 70 s each leg, each position, totaling 14 min Week 6: 3 × 70 s each leg, each position, totaling 14 min
Cloak et al., 2013 [23]	6 weeks twice a week	Frequency: 30 Hz (1 & 2 weeks), 35 Hz (3 & 4 weeks), and 40 Hz (5 & 6 weeks).	Vibrosphere (ProMedvi)	4 exercises: standing on 1 leg; heel raises on 1 leg; single-leg step-ups; single-leg straight-leg deadlift.	Week 1: 2 × 45 s each leg, each position, totaling 10 min Week 2: 2 × 45 s each leg, each position, totaling 10 min Week 3: 2 × 60 s each leg, each position, totaling 12 min Week 4: 2 × 60 s each leg, each position, totaling 12 min Week 5: 2 × 75 s each leg, each position, totaling 14 min Week 6: 2 × 75 s each leg, each position, totaling 14 min
Jeong et al., 2017 [36]	6 weeks 3 times per week	Frequency: 5 to 25 Hz and Amplitude: 3 to 6 mm. WBVE: 30 min per day NMT: 30 min per day = 5 min of warm-up exercise, 20 min of the main exercises, and 5 min of cool-down exercise.	Alternating vertical and horizontal vibration modes were used Wellengang START (Wellengang GmbH, Bayern, Germany)	WBVE: Weeks 1, 2, & 3: one-legged stance, cross-legged sway, runner’s pose, catch and throw a volleyball against a wall. Weeks 4, 5, & 6: one-legged stance with eyes shut, cross-legged sway with an elastic resistance band attached to the ankle, runner’s pose with single leg heel raises, catch and throw a tennis ball against a wall. NMT: exercises performed—one-leg sideways jumps, agility training, vertical jumps, and one-leg figure-eight jumps.	The WBVE consisted of a total of 30 min including 5 min exercise and 1 min rest of a session
Sierra-Gusmán et al., 2017 [40]	6 weeks 3 days a week (48 h between sessions)	Frequency: 30 Hz (weeks 1 & 2), 35 Hz (weeks 3 & 4), & 40 Hz (weeks 5 & 6), was also increased by 5 Hz every 2 weeks. Amplitude: increased from 2 mm (week 1) to 4 mm (weeks 2,3,4,5 & 6) after the first week; it was then maintained at 4 mm for the remainder of the study.	Fitvibe Excel Pro vibration platform (Fitvibe, Bilzen, Belgium)	Weeks 1, 2, & 3: one-legged stance; cross-legged sway; runner’s pose; catch and throw a volleyball against the wall. Weeks 4,5 & 6: one-legged stance with eyes shut; cross-legged sway with an elastic resistance band attached to the ankle; runner’s pose with single-leg heel raises; catch and throw a volleyball against the wall. Exercises performed with the BOSU^®^ on the VP.	The training program consisted of 3 series of 4 exercises of 45 s, with 45 s rest between exercises. Total of 18 min.
Sierra Gusmán et al., 2018 [51]	6-week balance-training protocol; 3 days each week (With 48 h between sessions)	Frequency: 30 Hz (weeks 1 & 2), 35 Hz (weeks 3 & 4), and 40 Hz (weeks 5 & 6), was also increased by 5 Hz every 2 weeks. Amplitude: increased from 2 mm (week 1) to 4 mm (weeks 2, 3, 4, 5, & 6) after the first week and then maintained for the remainder of the study. The level of difficulty of all exercises increased after 3 weeks.	Excel Pro vibration platform (Fitvibe, Bilzen, Belgium).	Weeks 1,2 & 3: legged stance; cross-legged sway; runner’s pose; catch and throw a volleyball against the wall. Weeks 4,5 & 6: legged stance with eyes shut; cross-legged sway with an elastic resistance band attached to the ankle; runner’s pose with single-legged heel raises; catch and throw a tennis ball against the wall NVIB group: trained with the BOSU on the floor; VIB group: trained on VP.	The training program consisted of 3 series of 4 exercises of 45 s, with 45 s rest between exercises. Total of 18 min.
Chang et al., 2021 [26]	6-week training programs (3 times per week)	Frequency and amplitude of 5 Hz and 3 mm.	Oscillating Vibration platform (AIBI Power Shaper, AIBI Fitness, Singapore)	The main exercise comprised three exercise movements: a double-leg stance, a one-legged stance, and a tandem stance.	5 min warm-up exercise, a 20 min main exercise, and a 5 min cool-down exercise. Weeks 1–3: 4 sets of 45 s exercises with a 40 s rest interval between exercises. Weeks 4–6: 5 sets of 45 s exercises with a 30 s rest interval between exercises.
Kruatiwa et al., 2021 [24]	6 weeks training (3 sessions a week) 18 sessions	Frequency and amplitude parameters of 30 Hz, 4 mm, 40 Hz, 4 mm, and 50 Hz, 8 mm. The intensity gradually increased every 2 weeks.	Power plate Pro5 silver (Power Plate International Ltd.).	The participants jointly performed single-leg static balance on a power plate. The protocol consisted of an unstable ankle in two poses: including single-leg heel raise and a 60 [degrees] single-leg squat.	Each pose with 15 s rest between sets in a bilateral pose training session. A total of 20 min.

WBVE: whole-body vibration exercise; NMT: neuromuscular training.

## Data Availability

The datasets used and/or analyzed during the current study are available from the corresponding author upon reasonable request.

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
