# Peer review of "Effects of Whole-Body Vibration Exercise on Athletes with Ankle Instability: A Systematic Review"

_ijerph, 2023, doi:10.3390/ijerph20054522_

Round 1

Reviewer 1 Report

The current systematic review was conducted on electronic searches in Pub-36 med, Cochrane Library, Embase, Scopus, Science Direct, Web of Science, Allied Health Literature 37 (CINAHL) and Academic Search Premier (ASP) (EBSCO) databases in September 2022. to assess effects of whole-body vibra-35 tion exercise (WBVE) in athletes with CAI. Chronic ankle instability (CAI) is a common condition following an ankle sprain. The study on effects of whole body vibration (WBV) and shoe with an unstable surface training on balance, functional performance, strength, joint position sense in people with CAI are common utility.

Research has shown that the surface of artificial turf soccer fields has caused more injuries than the natural turf surface in amateur footballers ( Pietro, M., Rocco, S., Felice, S., Madonna, G., & Filomena, M. (2020). Soccer fields in synthetic and natural grass: A comparative study on muscular injuries of the lower limb. Sport Science, 14, 7-12). This reference can you added in introduction to strengthen the topic. Authors should describe date when they conduct an online survey..

Author Response

REVIEWER 1

Review Report Form

Open Review

English language and style

( ) English very difficult to understand/incomprehensible
( ) Extensive editing of English language and style required
( ) Moderate English changes required
( ) English language and style are fine/minor spell check required
(x) I don't feel qualified to judge about the English language and style

Yes

Can be improved

Must be improved

Not applicable

Does the introduction provide sufficient background and include all relevant references?

(x)

( )

( )

( )

Are all the cited references relevant to the research?

(x)

( )

( )

( )

Is the research design appropriate?

(x)

( )

( )

( )

Are the methods adequately described?

(x)

( )

( )

( )

Are the results clearly presented?

(x)

( )

( )

( )

Are the conclusions supported by the results?

(x)

( )

( )

( )

Comments and Suggestions for Authors

The current systematic review was conducted on electronic searches in Pub-36 med, Cochrane Library, Embase, Scopus, Science Direct, Web of Science, Allied Health Literature 37 (CINAHL) and Academic Search Premier (ASP) (EBSCO) databases in September 2022. to assess effects of whole-body vibra-35 tion exercise (WBVE) in athletes with CAI. Chronic ankle instability (CAI) is a common condition following an ankle sprain. The study on effects of whole body vibration (WBV) and shoe with an unstable surface training on balance, functional performance, strength, joint position sense in people with CAI are common utility.

 Thank you for your comments and considerations.

Research has shown that the surface of artificial turf soccer fields has caused more injuries than the natural turf surface in amateur footballers ( Pietro, M., Rocco, S., Felice, S., Madonna, G., & Filomena, M. (2020). Soccer fields in synthetic and natural grass: A comparative study on muscular injuries of the lower limb. Sport Science, 14, 7-12). This reference can you added in introduction to strengthen the topic.

Thank you. We agree and added this reference in introduction.

Authors should describe date when they conduct an online survey.

We added this information. 

Submission Date

09 January 2023

Date of this review

16 Feb 2023 13:46:53

Reviewer 2 Report

Review. 

Title: Sufficient. 

Authors: Author list is quite long – please explain why. Please also check if all authors qualify  as authors according to guidelines. 

Abstract: Well written. 

Introduction:

A very informative introduction. Please put some more focus on anatomy of ankle instability in this section and make the differences and equalities between FAI and CAI clear. Please also explain for reader not knowing VP if the athlethes just stand on the plate or do exercises here. 

Please provide as last part of the introduction hypothesis and a clear aim of the study. 

Methods: 

Very clear and reliable. Please only add when referring Endnote also the Company and Country, e.g. Endnote (Clarivate Analystics, London, United Kingdeom). 

Results: 

L150: why were the studies excluded?

Please try to shorten the results. 

Figures: Please edit figure 1: provide a clear figure. The boxes of Identification/Screening/Inclusion are not in one line. The first box is not clear: I would suggest writing: “Reports identified from (n=122): pubmed n=xx, Cochrane-….. “) You referred here to “databases”  - what do you mean with this? Pubmed is as well a database…

Discussion: please add the perspectives more fluently in the text and not as new “subheading”. Please also add some references on VP in normal patients with ankle instability. 

Author Response

REVIEWER 2

Review Report Form

Open Review

English language and style

( ) English very difficult to understand/incomprehensible
( ) Extensive editing of English language and style required
( ) Moderate English changes required
(x) English language and style are fine/minor spell check required
( ) I don't feel qualified to judge about the English language and style

Yes

Can be improved

Must be improved

Not applicable

Does the introduction provide sufficient background and include all relevant references?

( )

(x)

( )

( )

Are all the cited references relevant to the research?

(x)

( )

( )

( )

Is the research design appropriate?

(x)

( )

( )

( )

Are the methods adequately described?

(x)

( )

( )

( )

Are the results clearly presented?

( )

(x)

( )

( )

Are the conclusions supported by the results?

( )

(x)

( )

( )

Comments and Suggestions for Authors Review. 

Thank you for your comments and considerations.

Title: Sufficient. 

Authors: Author list is quite long – please explain why. Please also check if all authors qualify as authors according to guidelines. 

Our scientific group has a collaboration of graduate and undergraduate students that study the use of systemic vibratory therapy in individuals with various clinical conditions, and even in healthy individuals. These students always are discussing our findings and the manuscripts that are ongoing. Moreover, we have the collaboration of colleagues in some countries, which helps us in the discussion of the findings and manuscripts to have the best quality of our research. In addition, these colleagues also help with the corrections of the English language.

Abstract: Well written. 

Introduction:

A very informative introduction. Please put some more focus on anatomy of ankle instability in this section and make the differences and equalities between FAI and CAI clear.

We agree and we thank you for this comment. We added this information about the anatomy and the differences between functional and mechanical instability in introduction.

Please also explain for reader not knowing VP if the athlethes just stand on the plate or do exercises here. Please provide as last part of the introduction hypothesis and a clear aim of the study. 

We agree, and we added this informations according it was requested.

Methods: 

Very clear and reliable. Please only add when referring Endnote also the Company and Country, e.g. Endnote (Clarivate Analystics, London, United Kingdom).

We agree, and we added this information. 

Results: 

L150: why were the studies excluded?

We agree, and we added this information.

Please try to shorten the results. 

Figures: Please edit figure 1: provide a clear figure. The boxes of Identification/Screening/Inclusion are not in one line. The first box is not clear: I would suggest writing: “Reports identified from (n=122): pubmed n=xx, Cochrane-….. “) You referred here to “databases”  - what do you mean with this? Pubmed is as well a database…

We agree, and we edit the Figure 1.

Discussion: please add the perspectives more fluently in the text and not as new “subheading”.

We agree, and we inserted the facts and perspectives in the text.

Please also add some references on VP in normal patients with ankle instability. 

We used references with WBVE and individuals with chronic ankle instability in the discussion.

References:

  1. Rendos, N.K.; Jun, H.P.; Pickett, N.M.; Lew Feirman, K.; Harriell, K.; Lee, S.Y.; Signorile, J.F. Acute effects of whole body vibration on balance in persons with and without chronic ankle instability. Research in sports medicine (Print) 2017, 25, 391-407, doi:10.1080/15438627.2017.1365299.
  2. Shamseddini Sofla, F.; Hadadi, M.; Rezaei, I.; Azhdari, N.; Sobhani, S. The effect of the combination of whole body vibration and shoe with an unstable surface in chronic ankle instability treatment: a randomized clinical trial. BMC Sports Science, Medicine and Rehabilitation 2021, 13, doi:10.1186/s13102-021-00256-6.
  3. Tohidast, S.A.; Bagheri, R.; Safavi-Farokhi, Z.; Khaleghi Hashemian, M.; Delkhosh, C.T. The Effects of Acute and Long-Term Whole-Body Vibration Training on the Postural Control During Cognitive Task in Patients With Chronic Ankle Instability. Journal of sport rehabilitation 2021, 30, 1121-1128, doi:10.1123/jsr.2021-0034.
  4. Adelman, D.; Pamukoff, D.; Goto, S.; Guskiewicz, K.; Ross, S.; Blackburn, T. Acute Effects of Whole Body Vibration on Dynamic Postural Control and Muscle Activity in Individuals With Chronic Ankle Instability. Athletic Training and Sports Health Care 2016, 8, 63-69, doi:10.3928/19425864-20160204-01.
  5. Hartsell, H.D.; Spaulding, S.J. Eccentric/concentric ratios at selected velocities for the invertor and evertor muscles of the chronically unstable ankle. British journal of sports medicine 1999, 33, 255-258, doi:10.1136/bjsm.33.4.255.
  6. Otzel, D.M.; Hass, C.J.; Wikstrom, E.A.; Bishop, M.D.; Borsa, P.A.; Tillman, M.D. Motoneuron Function Does not Change Following Whole-Body Vibration in Individuals With Chronic Ankle Instability. Journal of sport rehabilitation 2019, 28, 614-622, doi:10.1123/jsr.2017-0364.

Submission Date

09 January 2023

Date of this review

25 Jan 2023 11:39:17

Reviewer 3 Report

This study is very interesting, well written and really clear and easy to read. There is one sentence I am not sure if it is a typo, it needs to be rephrased. Page 7, line 222: Biodex Balance System Test was used to ankle balance to assessed using the BBS.

Author Response

REVIEWER 3

Review Report Form

Open Review

English language and style

( ) English very difficult to understand/incomprehensible
( ) Extensive editing of English language and style required
( ) Moderate English changes required
(x) English language and style are fine/minor spell check required
( ) I don't feel qualified to judge about the English language and style

Yes

Can be improved

Must be improved

Not applicable

Does the introduction provide sufficient background and include all relevant references?

(x)

( )

( )

( )

Are all the cited references relevant to the research?

(x)

( )

( )

( )

Is the research design appropriate?

(x)

( )

( )

( )

Are the methods adequately described?

(x)

( )

( )

( )

Are the results clearly presented?

(x)

( )

( )

( )

Are the conclusions supported by the results?

(x)

( )

( )

( )

Comments and Suggestions for Authors

This study is very interesting, well written and really clear and easy to read.

Thank you for your comments and considerations.

There is one sentence I am not sure if it is a typo, it needs to be rephrased. Page 7, line 222: Biodex Balance System Test was used to ankle balance to assessed using the BBS.

Thank you for this observation. We correct this sentence in the manuscript.

‘‘Biodex Balance System Test (BBS) was used to assessed ankle balance...’’

Submission Date

09 January 2023

Date of this review

17 Feb 2023 21:35:32

Reviewer 4 Report

Thanks for the opportunity to review this paper. This study aimed to analyze the effects of whole-body vibration exercise in athletes with ankle instability through a systematic review.

The authors explore the gap and synthesize the evidence of interventions that used the vibrating platform in athletes with ankle instability.

First, I express my positive evaluation of the manuscript, which contains results that justify the publication in IJERPH. However, I have concerns that led to my decision to major reviews.

The main concerns are about the Methods and Results.

Below, I present the main reasons for my decision.

Title

Evidence about: I don't think it's necessary. I suggest removing

Abstract

The objective is confusing and seems redundant. Furthermore, it does not refer to athletes, nor that the comparison is between amateur and elite athletes.

Protocol study registration: PROSPERO (CRD42020203295).

I suggest updating the protocol in PROSPERO for completed and unpublished reviews.

Are there any differences between the protocol and the manuscript (Protocol P: individuals with ankle instability vs. Manuscript: athletes with CAI)?

Introduction

The introduction presents a background the prevalence of ankle injuries in athletes, associated with negative outcomes and the development of chronic ankle instability (CAI) or Functional Ankle Instability (FAI) and the possible positive effects of using the vibrating platform.

However, despite pointing out a blunt gap, this SR does not answer the “what is the most effective type of intervention”? For this, authors should be conducted a network.

Additionally, I suggest a more in-depth state-of-the-art, in particular, the results of the RS pointed out (Tan, J.; Wu, X.; Clark, C.C.T.; Barton, V.; Chen, S.; Liu, S.; Zhou, X .; Xu, C.; Ma, T.; Qi, B et al. The effect of whole-body vibration on sensorimotor deficits in people with chronic ankle instability: A systematic review and meta-analysis. Clinical rehabilitation 2022, 36, 1016-1031, doi:10.1177/02692155221095651).

Materials and Methods

Add the checklist as a supplementary document.

Was the method designed for a systematic review without meta-analysis or defined as a posteriori due to the heterogeneity of the studies? The protocol, authors state: There will be no analysis of data statistics, as the purpose of this review is not to develop a meta-analysis. In the discussion, they state: Finally, it was not possible to conduct a meta-analysis -analysis, considering the heterogeneity in intervention protocols and study samples.

Please clarify.

Search Strategy

Scopus and Science Direct are redundant with Embase. The number of duplicates was inflated.

The search in gray literature and references of the selected studies was not carried out. Why was this decision made?

In which field was the search performed?

Was there a filter?

Why not use mesh terms?

I suggest redoing and presenting the search considering these questions as a supplementary document in at least three different bases.

Eligibility Criteria

Redundant: SR included only randomized clinical trials. Then RS and MA are automatically not included). It is necessary to identify, from those included studies, what led to the exclusion in a subsequent step.

Methodological Quality and Risk of Bias

Why use the PEDro and ROB scales? PEDro also assesses the risk of bias.

I suggest opting for the updated version of the Cochrane Collaboration tool (ROB-II)

My biggest concern is about the studies included with recreational athletes. Recreational is too broad a spectrum to be included because your eligibility criteria are athletes.

Results

Figure 1. What does “Records marked as ineligible by automation tools (n = 14)” mean? clarify

Reports sought for retrieval (n = 0)

Reports not retrieved (n = 0)

Is this correct? A screening step?

What was the total number of athletes combined, total men and women, age range, sports

Table 1

Also indicate the study reference in the table

typo error:

Cloak et al. 2010 - 163.6±7.3em (Demographics)

What is the need to report the “aim”? what can be inferred from this parameter?

Variables measures

I suggest stratifying into.

Instruments, tests, outcomes and summarize separately

Results Still text-heavy and polluted. It is possible to further synthesize the results.

The age of the participants is omitted in some studies

Chang et al. 2021 groups

Kruatiwa et al. 2021 level of evidence?

Table 2

Data on adherence, adverse effects, and abandonment if reported, can compose this table

Discussion

Pg 13 L 286: What about the risk of bias?

Pg 14 L 345: “efficient exercise”. However, most of the studies have a small effect size.

Pg 14 L 347 I did not identify any results that support “greater safety, greater safety and consequently, less chance of recurrence of the lesion that promoted instability” was this extracted?

Figure 1. What does “Records marked as ineligible by automation tools (n = 14)” mean? clarify

Reports sought for retrieval (n = 0)

Reports not retrieved (n = 0)

Is this correct? A screening step?

What was the total number of athletes combined, men and women, age range, sports?

Table 1

Also indicate the study reference in the table

typo error:

Cloak et al. 2010 - 163.6±7.3em (Demographics)

What is the need to report the “aim”? What can inferred from this parameter?

Variables measures.

I suggest stratifying:

Instruments, tests, outcomes and summarize separately.

Results Still text-heavy and polluted. It is possible synthesize the results.

Was omitted in some studies the age of the participant?

Chang et al. 2021 groups

Kruatiwa et al. 2021 level of evidence?

Table 2

adherence, adverse effects, and abandonment when reported can compose this table.

Discussion

Pg 13 L 286: What about the risk of bias?

Pg 14 L 345: “efficient exercise.” However, most of the studies have a small effect size.

Pg 14 L 347 I did not identify any results that support “greater safety, greater safety and consequently, less chance of recurrence of the lesion that promoted instability” was this extracted?

Author Response

REVIEWER 4

Review Report Form

Open Review

English language and style

( ) English very difficult to understand/incomprehensible
( ) Extensive editing of English language and style required
( ) Moderate English changes required
(x) English language and style are fine/minor spell check required
( ) I don't feel qualified to judge about the English language and style

Comments and Suggestions for Authors

Thanks for the opportunity to review this paper. This study aimed to analyze the effects of whole-body vibration exercise in athletes with ankle instability through a systematic review.

The authors explore the gap and synthesize the evidence of interventions that used the vibrating platform in athletes with ankle instability.

First, I express my positive evaluation of the manuscript, which contains results that justify the publication in IJERPH. However, I have concerns that led to my decision to major reviews.

The main concerns are about the Methods and Results.

Below, I present the main reasons for my decision.

Thank you for your comments and considerations.

Title

Evidence about: I don't think it's necessary. I suggest removing.

We removed "Evidence about" from title.
New title: “Effects of Whole-Body Vibration Exercise in Athletes with Ankle Instability: A Systematic Review”

Abstract

The objective is confusing and seems redundant. Furthermore, it does not refer to athletes, nor that the comparison is between amateur and elite athletes.

The idea of this review is to preset the possibility of using an efficient exercise program (systemic vibratory therapy) to assist the individualas in  dysfunctions promoted by CAI, and useful to stimulate the physical-functional performance of athletes, as well as in the return to sports practice. So, the aim of this study was is to present a systematic review to analyze  effects of WBVE in athletes with CAI, independently if amateur or elite athletes.

Protocol study registration: PROSPERO (CRD42020203295).

I suggest updating the protocol in PROSPERO for completed and unpublished reviews.

Thank you. We update the protocol in PROSPERO for completed and unpublished reviews.

Are there any differences between the protocol and the manuscript (Protocol P: individuals with ankle instability vs. Manuscript: athletes with CAI)?

With the protocol update in PROSPERO, we provide a brief description of the changes made to the manuscript, including the PICOS strategy, and eligibility criteria. However, it is important to consider that the protocol registration was carried out the register was realized in the initial phase of the review. During the process, the review was directed to athletes with CAI, but all other aspects of the revision remained the same.

Introduction

The introduction presents a background the prevalence of ankle injuries in athletes, associated with negative outcomes and the development of chronic ankle instability (CAI) or Functional Ankle Instability (FAI) and the possible positive effects of using the vibrating platform.

However, despite pointing out a blunt gap, this SR does not answer the “what is the most effective type of intervention”? For this, authors should be conducted a network.

The aim of this review is not to answer "which type of intervention is most effective", but rather to highlight the potential of systemic vibration therapy, based on whole-body vibration exercises, as an exercise alternative capable of promoting benefits for CAI population.

Additionally, I suggest a more in-depth state-of-the-art, in particular, the results of the RS pointed out (Tan, J.; Wu, X.; Clark, C.C.T.; Barton, V.; Chen, S.; Liu, S.; Zhou, X .; Xu, C.; Ma, T.; Qi, B et al. The effect of whole-body vibration on sensorimotor deficits in people with chronic ankle instability: A systematic review and meta-analysis. Clinical rehabilitation 2022, 36, 1016-1031, doi:10.1177/02692155221095651).

We agree, and we add this information in the introduction as follows:
‘’
Currently, there is only one systematic review about WBVE and sensorimotor effects, that supports the use of WBVE to improve sensorimotor deficits involving balance, strength, joint position sense, and muscle activity, but it was with CAI individuals, not athletes’’.

Materials and Methods

Add the checklist as a supplementary document.

We added the checklist PRISMA as a supplementary document.

Was the method designed for a systematic review without meta-analysis or defined as a posteriori due to the heterogeneity of the studies? The protocol, authors state: There will be no analysis of data statistics, as the purpose of this review is not to develop a meta-analysis. In the discussion, they state: Finally, it was not possible to conduct a meta-analysis -analysis, considering the heterogeneity in intervention protocols and study samples.

Please clarify.

We agree, and we have deleted this information from the discussion. The method was designed without meta-analysis.

Search Strategy

Scopus and Science Direct are redundant with Embase. The number of duplicates was inflated. Was there a filter?

We did not use any filters in the searches that  were carried out in eight databases, precisely to give greater credibility to the search, and the authors' desired to find the maximum number of available records on the subject of the review.

The search in gray literature and references of the selected studies was not carried out. Why was this decision made? document in at least three different bases.

We realized a hand search in the gray literature and at studies references, and we added that this search was performed in the text and in the Figure 1.

In which field was the search performed?

We performed simple and advanced searches.

Why not use mesh terms?
We searched for MeSH terms, but there were no specific terms available according to the research objectives to be used in the searches.

I suggest redoing and presenting the search considering these questions as a supplementary

We updated the search in the databases.

Eligibility Criteria

Redundant: SR included only randomized clinical trials. Then RS and MA are automatically not included). It is necessary to identify, from those included studies, what led to the exclusion in a subsequent step.

We agree, and we deleted the RS and MA if exclusion criteria. Additionally, we added information about the reason for the exclusion of the studies in the text and in the Figure 1.

Methodological Quality and Risk of Bias

Why use the PEDro and ROB scales? PEDro also assesses the risk of bias.

The PEDro can also be considered an analysis of methodological quality, in addition to the risk of bias. And because it is a research within the field of physiotherapy, we were interested in using it.

I suggest opting for the updated version of the Cochrane Collaboration tool (ROB-II)

It would be interesting, thanks for the consideration. But, we opted for this version because it is also used and also includes the necessary assessments regarding the evaluation of the risk of bias. Additionally, the information is complemented by the PEDro scale.

My biggest concern is about the studies included with recreational athletes. Recreational is too broad a spectrum to be included because your eligibility criteria are athletes.

It would be interesting, and we are planing in the future to write another SR about this. But, when we selected athletes as the study population, we really didn't think about determining the level of the athlete. Our intention was to study individuals who practice a certain sport modality, that is, athletes, regardless of whether they are professional or recreational. We are dealing with studies with individuals who practice the same sport, at the competition level of that modality, and these are always evaluated between them, which minimizes complications when we synthesize and compare the studies.

Results

Figure 1. What does “Records marked as ineligible by automation tools (n = 14)” mean? Clarify

We deleted this information.

Reports sought for retrieval (n = 0)/Reports not retrieved (n = 0)
Is this correct? A screening step?

We deleted this information.

What was the total number of athletes combined, total men and women, age range, sports

We added this information.

Table 1

Also indicate the study reference in the table

We added the studies references in the tables 1 and 2.

typo error:

Cloak et al. 2010 - 163.6±7.3em (Demographics)

We corrected this error as follows: Cloak et al. 2010 - 163.6±7.3cm (Demographics)

What is the need to report the “aim”? what can be inferred from this parameter?

We deleted this column.

Variables measures

I suggest stratifying into.

Instruments, tests, outcomes and summarize separately. Results Still text-heavy and polluted. It is possible to further synthesize the results.

We agree, and we synthesize the results in Table 1 in columns: instruments, measured variables and tests.

The age of the participants is omitted in some studies

We added information about age, height, and body mass in Table 1 for each study.

Chang et al. 2021 groups

We added information about the groups in Table 1.

Kruatiwa et al. 2021 level of evidence?

We added the level of evidence in Table 1.

Table 2

Data on adherence, adverse effects, and abandonment if reported, can compose this table

In general, these data  were not reported in the selected studies.

Discussion

Pg 13 L 286: What about the risk of bias?

We inserted information about the risk of bias, follow as:

‘‘ However, according to the risk of bias, the included clinical trials still lack detailed information on the methods used and have methodological errors that may compromise the internal validity.’’

Pg 14 L 345: “efficient exercise”. However, most of the studies have a small effect size.

It is an interesting comment. Based on the evidence presented in this systematic review, we suggested the WBVE as a possibility of exercise for this population, which can be considered efficient. Although the size of the effect, in general, all studies demonstrated clinical benefits and significant results of systemic vibratory therapy (SVT), as a Whole Body Vibration exercise (WBVE), reinforcing the idea of its use to assist in the dysfunctions promoted by CAI, and useful to stimulate the physical-functional performance of the athlete, and consequently these effects,  they are important to the return of the individuals to sports practice.

Pg 14 L 347 I did not identify any results that support “greater safety, greater safety and consequently, less chance of recurrence of the lesion that promoted instability” was this extracted?

It is an interesting comment, but the literature has been  shown that exercises on a vibrating platform are safe, feasible, and well-tolerated by patients with different disorders, with benefits associated with diverse parameters. [1]. Furthermore, the literature suggests exercises that increase static/dynamic balance and fatigue resistance, as well as WBVE, should be performed following an ankle injury to allow a safe return to sporting activity[2].

  1. Stania M., Juras G., SÅ‚omka K., Chmielewska D., Król P. The application of whole-body vibration in physiotherapy – a narrative review. Acta Physiologica Hungarica. . 2016;103(2):133–145. doi: 10.1556/036.103.2016.2.1.
  2. Cloak, R.; Nevill, A.M.; Clarke, F.; Day, S.; Wyon, M.A. Vibration training improves balance in unstable ankles. International journal of sports medicine 2010, 31, 894-900.

Submission Date

09 January 2023

Date of this review

24 Feb 2023 06:32:53
